# Drosophila TNFRs Grindelwald and Wengen bind Eiger with different affinities and promote distinct cellular functions

Valentina Palmerini[1], Silvia Monzani[1], Quentin Laurichesse[2,3], Rihab Loudhaief[2,3], Sara Mari[1], Valentina Cecatiello[1], Vincent Olieric [4], Sebastiano Pasqualato [1,5✉], Julien Colombani [2,3,5✉], Ditte S. Andersen[2,3,5✉] & Marina Mapelli [1,5✉]

The *Drosophila* tumour necrosis factor (TNF) ligand-receptor system consists of a unique ligand, Eiger (Egr), and two receptors, Grindelwald (Grnd) and Wengen (Wgn), and therefore provides a simple system for exploring the interplay between ligand and receptors, and the requirement for Grnd and Wgn in TNF/Egr-mediated processes. Here, we report the crystallographic structure of the extracellular domain (ECD) of Grnd in complex with Egr, a high-affinity hetero-hexameric assembly reminiscent of human TNF:TNFR complexes. We show that ectopic expression of Egr results in internalisation of Egr:Grnd complexes in vesicles, a step preceding and strictly required for Egr-induced apoptosis. We further demonstrate that Wgn binds Egr with much reduced affinity and is localised in intracellular vesicles that are distinct from those containing Egr:Grnd complexes. Altogether, our data provide insight into ligand-mediated activation of Grnd and suggest that distinct affinities of TNF ligands for their receptors promote different and non-redundant cellular functions.

[1] IEO, European Institute of Oncology IRCCS, Milan, Italy. [2] Department of Biology, Faculty of Science, University of Copenhagen, Copenhagen O, Denmark. [3] Novo Nordisk Foundation Center for Stem Cell Research, Faculty of Health and Medical University of Copenhagen, Copenhagen N, Denmark. [4] Paul Scherrer Institute, Villigen-PSI, Switzerland. [5] These authors contributed equally: Sebastiano Pasqualato, Julien Colombani, Ditte S. Andersen, Marina Mapelli. ✉email: sebastiano.pasqualato@ieo.it; julien.colombani@bio.ku.dk; ditte.andersen@bio.ku.dk; marina.mapelli@ieo.it

The family of tumour necrosis factors (TNFs) are implicated in diverse processes ranging from cell proliferation, differentiation and apoptosis to innate and adaptive immunity. While TNFs contribute to tissue homoeostasis and immunity, they are also notorious for their pathological roles in promoting tumour growth and inflammatory diseases. In mammals, the TNF-alpha superfamily is composed of 19 ligands and the TNF receptor (TNFR) superfamily of 29 related receptors[1]. Several TNF ligands can activate more than one TNFR and hence trigger distinct and sometimes opposing cellular responses. Due to functional redundancy and the complexity of the mammalian TNF ligand-receptor network, the mechanisms underlying selective activation of one TNFR over another remains poorly understood. *Drosophila* has a much simpler TNF:TNFR system with only one TNF ligand, Egr, and two TNFRs, Grnd and Wgn, and therefore provides an excellent model to explore the relationship between ligand and receptors and how this relates to the many and diverse functions attributed to Egr/TNF[2–5].

Members of the TNF-alpha superfamily are type II transmembrane proteins, with a well-conserved TNF-homology domain (THD) folding as a jelly-roll β-sandwich and responsible for ligand trimerization and receptor binding. Consistently, *Drosophila* Egr is a 409 residue glycoprotein coding for a single helix transmembrane (TM) domain and a C-terminal THD (Fig. 1A), that based on the structure of the *Spodoptera frugiperfda* (Sf9) ortholog is predicted to form trimers[6]. A diffusible form of Egr generated by cleavage of the THD from the TM helix by the TNFα-converting-enzyme (TACE) is found in the fly hemolymph, and accounts for short- and long-range non-autonomous Egr-mediated responses[5,7–9].

Mechanistically, TNF signalling is triggered by binding of TNF ligands to cognate surface TNFRs that generates an intracellular signalling cascade of second messengers driving either NF-kB-mediated proliferation or caspase-dependent apoptosis. Canonically, ligand engagement by surface TNFRs induces recruitment of adaptors molecules including FADD/TRADD proteins and Traf family proteins, which are the upstream cytoplasmic effectors of JNK signalling. Extensive studies on human TNFR activation revealed that ligand-mediated pro-apoptotic and anti-apoptotic signalling proceeds with different timings, promoted by the formation of molecularly and spatially distinct complexes that sequentially activate NF-KB and caspases[10]. Ligand engagement of TNFR1 and CD95 first induces the assembly of cell-surface complexes sufficient for JNK and NF-kB signalling, followed by internalization of the TNF/TNFR complexes that leads to full-activation and concomitant caspase stimulation.

The molecular events underlying Grnd and Wgn activation are just beginning to be understood. Wng was identified nearly two decades ago by mild sequence homology with cysteine-rich-domains (CRDs) of human TNFRs[11] and by interaction with Egr[4], although it seems to lack an intracellular death-domain or Traf-interacting-motif required for canonical JNK signalling. We recently identified a second *Drosophila* TNFR, Grnd, which interacts with both Egr and Traf2 and mediate Egr pro-apoptotic functions[5]. In spite of the very poor sequence homology with other TNFRs, we showed that Grnd harbours an extracellular CRD interacting with Egr, and intracellular motifs directly binding to Traf2 and to the Lin7 subunit of the apical polarity determinant, Crumbs[5]. Interestingly, N-linked glycosylation of Grnd has been reported to modulate remote JNK activation by decreasing the Grnd/Egr-binding affinity[9], although the molecular details remain poorly understood.

How Wng and Grnd cross-talk in transducing Egr signals in vivo is still largely unclear. For instance, it is still unclear whether Grnd and Wgn respond to different thresholds of Egr, are redundant and/or have distinct cellular or tissue-specific functions in Egr-mediated processes. Our previous data clearly show that, at least for some Egr-mediated processes, Grnd and Wgn are not redundant[5]. Thus, knocking down Grnd completely blocks Egr-induced apoptosis in the eye compound and the Egr-mediated metastatic behaviour of $Ras^{V12}/scrib^{-/-}$ tumours, whereas neither of these processes is suppressed in *wgn* null mutant animals. Grnd is expressed in larval wing and eye imaginal discs, where it localizes at the apical membrane[5]. Although previous reports suggested a role of Wgn in transducing Egr signalling in eye discs[4,12], its expression or compartmentalization in this tissue has not been studied. Hence, it remains unclear whether Wng and Grnd are both expressed in these structures, and if so whether they reside in functionally distinct complexes or cross-talk in transducing Egr signals.

Here we report the biochemical characterization of the molecular complexes of Egr with Grnd and Wgn, showing that both receptors form hetero-hexamers with trimeric Egr ligands, although with much different affinities. Moreover, Grnd and Wgn localizes to distinct cellular compartments in wing discs, suggesting that the two receptors serve diverse and non-redundant cellular functions.

## Results

**Grnd and Wgn have different affinities for Egr**. To start exploring the nature of the interaction of Egr/TNF with its receptors, Grnd and Wgn, we first reconstituted the complexes using the entire extracellular portion of Egr comprising residues 146–409, which encompasses the C-terminal TNF-homology domain and corresponds to portion of the Egr found in the hemolymph after cleavage by TACE proteinase[8,11] (Fig. 1A), and the extracellular fragments of Grnd and Wgn-containing single cysteine-rich domains (CRDs), encompassing residues 30–97 and 78–201, respectively (Fig. 1A). When loaded on a size-exclusion column in equimolar amounts at 30 µM concentration, $Grnd^{30-97}$ and $Egr^{146-409}$ eluted in a single peak at a molecular weight corresponding to the 158 kDa molecular weight marker, indicating that they assemble in a high-stoichiometry complex (Fig. 1B). Under the same conditions, $Egr^{146-409}$ and $Wgn^{78-201}$ did not interact (Supplementary Fig. S1A, B). However, $Egr^{146-409}$ co-eluted with $Wgn^{78-201}$ when incubated at 200 µM concentration in a 1:4 ratio with the receptor, in fractions around the 158 kDa molecular weight marker (Fig. 1C). Notably, $Egr^{146-409}$ in isolation elutes near the 158 KDa marker as well, in spite of being an homotrimer of about 92 KDa (Fig. 1D), likely because it consists of a globular TNF-homology-domain preceded by a 124-residue long unstructured region that might affect its hydrodynamic properties. To better define the stoichiometry of the interactions, we performed static light-scattering (SLS) analyses which revealed that $Egr^{146-409}$ is trimeric and engages with monomeric $Grnd^{30-97}$ and $Wgn^{78-201}$ in hetero-hexameric complexes (Fig. 1D). In agreement with the size-exclusion chromatography (SEC) experiments, quantification of the binding strength between the ligand and the receptors revealed that Egr associates with Grnd with nanomolar affinity ($K_D$ around 30 nM), and with Wgn with low micromolar affinity ($K_D$ in the order of 12 µM) (Fig. 1E). Collectively these results indicate that the trimeric ECD of Egr interacts directly with the CRDs of the two *Drosophila* TNFRs, Grnd and Wgn, forming hetero-hexameric complexes with an affinity that is three orders of magnitude higher for Grnd than for Wgn, suggesting that the two receptors could rely on different ligand thresholds for activation.

**Structure of the extracellular domain of Grnd**. To uncover the organizational principles of the Grnd:Egr interaction, we first determined the crystallographic structure of the extracellular

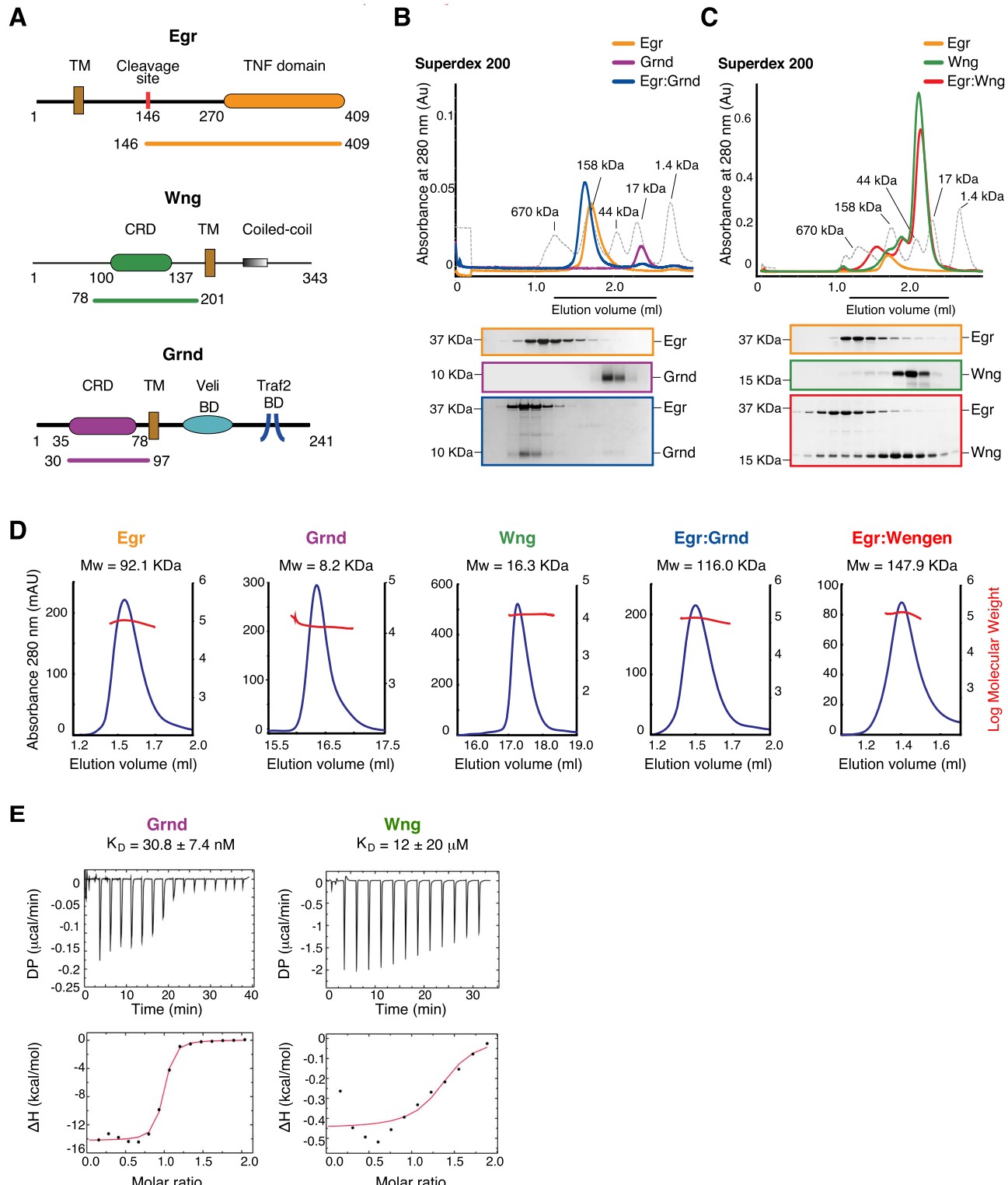

portion of Grnd. The Grnd[30–97] constructs proved refractory to the first crystallization attempts. Therefore, we further trimmed its boundaries by limited proteolysis with trypsin to a shorter fragment, which was assigned by mass spectrometry to Grnd[30–81] (Supplementary Fig. S2A). The proteolytically defined Grnd[30–81] readily yielded crystals diffracting to 0.92 Å resolution. The structure was solved by S-SAD, and refined to a $R_{\text{free}}$ of 0.16 and good stereochemistry (Table 1 and Supplementary Fig. S2B). The final model includes residues 34–81 of Grnd, which we will refer to as Grnd-ECD hereon. The structure of Grnd-ECD consists of

an N-terminal β-hairpin followed by two C-terminal antiparallel α-helices, with the 8 cysteines engaged in four disulfide bonds (Fig. 2A, B), with a topology that is reminiscent of the CRD fold seen in structurally known TNFRs. All known TNFR CRDs are organized in two modules, with distinctive tertiary fold and consensus sequence[13]. The N-terminal module of Grnd spanning residues 35–54 conforms to an X2 module, composed of β-hairpin stabilized by two intercalated disulfide bonds, with a topology found also in human BAFFR[14,15] (Fig. 2C and Supplementary Fig. S2C). The C-terminus of the Grnd-CRD folds in a

**Fig. 1 Egr forms hetero-hexamers with both Grnd and Wgn. A** Top: Egr domain structure comprising an N-terminal intracellular region, a transmembrane (TM) helix, and an extracellular portion with the TNF domain. The cleavage site recognized by TACE is indicated as a red bar. Middle: Domain structure of Wgn with an N-terminal cysteine-rich domain (CRD), a TM and an intracellular region with a coiled-coil. Bottom: Domain structure of Grnd with an N-terminal extracellular region containing a CRD, followed by a TM domain, and an intracellular region containing the Veli and Traf2 binding domains. **B**, **C** SEC elution profile of the complex formed between Egr[146-409] and Grnd[30-97] mixed at 30 μM equimolar concentration, and Egr[146-409] and Wgn[78-201] mixed at 200 μM:800 μM concentrations, with Coomassie-stained SDS-PAGE of the eluted fractions corresponding to the horizontal black bar. The elution profile of globular markers is reported as a dashed grey line. Individual runs of Egr, Grnd and Wgn are shown for comparison. Results of SEC runs and SDS-PAGES were confirmed by successful replicate experiments. **D** Static-Light-Scattering profiles of Egr[146-409], Grnd[30-97], Wgn[78-201], and the two complexes formed by Egr[146-409]:Grnd[30-97] and Egr[146-409]:Wgn[78-201]. The UV absorbance trace is shown in blue (left axis) and the measured MW in red (right axis). **E** ITC measurements of the binding affinity between Egr[146-409] and Grnd[30-97] (left) and Egr[146-409] and Wgn[78-201](right). The $K_D$ is reported as mean ± error fitting of the ITC data with the isotherm (black line). Both reactions are exothermic. Uncropped images of SDS-PAGE gels and immunoblots are provided in Supplementary Fig. 6.

---

**Table 1 Data collection and refinement statistics of the Grnd structure.**

| | Grnd | |
|---|---|---|
| *Data collection* | Native PXIII-SLS | S-SAD PXIII-SLS |
| Space group | $P4_12_12$ | $P4_12_12$ |
| Wavelength | 0.93 Å | 2.07 Å |
| Cell dimensions | | |
| $a$, $b$, $c$ (Å) | 31.48, 31.48, 97.76 | 31.45, 31.45, 97.75 |
| $\alpha$, $\beta$, $\gamma$ (°) | 90.0, 90.0, 90.0 | 90.0, 90.0, 90.0 |
| Resolution (Å) | 27.97-0.93 (0.94-0.93)[a] | 48.88-1.92 (5.21-1.92)[a] |
| $R_{sym}$ or $R_{merge}$ | 0.064 (0.412) | 0.073 (0.04) |
| $CC_{1/2}$ | 1.00 (0.85) | 1.00 (1.00) |
| $I/\sigma I$ | 21.5 (2.4) | 163.4 (11.8) |
| Completeness (%) | 98.2 (83.8) | 96.4 (43.2) |
| Unique reflections | 33869 (1382) | 4015 (89) |
| Multiplicity | 10.6 (3) | 221.8 (5.7) |
| *Refinement* | | |
| Resolution (Å) | 29.97-0.93 | |
| $R_{work}$/$R_{free}$ | 0.130/0.157 | |
| No. atoms | | |
| Protein | 404 | |
| Water | 83 | |
| B-factors protein/water (Å$^2$) | 11.31/35.17 | |
| R.m.s. deviations | | |
| Bond lengths (Å) | 0.016 | |
| Bond angles (°) | 2.37 | |
| Ramachandran values | | |
| Favoured (%) | 100 | |
| Allowed (%) | 0 | |
| Outliers (%) | 0 | |

[a]Values in parentheses are for the highest-resolution shell.

---

C2 helix-loop-helix module, organized in a nested pattern of disulfide bridges, according to the consensus sequence Cys57-x$_3$-Cys61-x$_6$-x$_5$-Cys73-x$_3$-Cys77 (where x is any given amino acids, and the subscript n indicates the number of residues). Interestingly, C2 modules are also present in the CRD of human Fn14 and in the fourth CRD of TNFRI[16,17]. Although X2 and C2 modules are common features of TNFRs, Grnd-ECD is the first example of a TNFR fold combining the two together, with the side-chain of Phe46 fitting in between the two (Fig. 2A). In spite of the poor sequence similarity, the overall topology of Grnd resembles that of human Fn14 and BCMA[14,18], and the second CRD of TACI[19], although it superposes poorly with the extracellular domains of these receptors (Supplementary Fig. S2D–G). Most notably, these receptors use both modules to interact with their ligands, with major contacts from the β1–β2-hairpin tip

harbouring an Asp-x-Leu/Ser-x-Asp-Leu motif, that is not present in Grnd (Supplementary Fig. S2C).

**Architecture of the Egr:Grnd hetero-hexameric complex.** Both Grnd and Wgn have been implicated in Egr-mediated signal transduction, although the molecular mechanisms underlying these activities and their specificity remain largely unclear. To shed light on the mechanism of Grnd activation by Egr, we set out to determine the crystallographic structure of the complex. To obtain diffraction-quality crystals, the Grnd[30-97]:Egr[146-409] sample was subjected to limited proteolysis, yielding shorter constructs assigned by mass spectrometry to Egr[269-409] (Fig. 3A) and Grnd[30-81], that still form complexes in solution according to SEC analysis (Supplementary Fig. S3A, B). The Egr fragment obtained by trypsinization corresponds to the TNF-homology domain, sharing good conservation with TNF ligands that are known to associate with single-CRD TNFRs including TALL1, April, and TWEAK[14,19] (Supplementary Fig. S3D). Grnd[30-81]:Egr[269-409] crystals diffracted to a resolution of 2.02 Å and belonged to the *P*321 space group, reflecting the threefold symmetry axis observed in the complex. The structure was solved by molecular replacement using as search model for the Egr sequence threaded on the EDA-A2 TNF domain (PDB-ID 1RJ8), and later adding copies of the Grnd-ECD structure previously solved. The structure was refined to $R_{free}$ of 0.21 and $R_{work}$ of 0.19 with good stereochemistry (Table 2 and Supplementary Fig. S3C). The final model includes residues 269–409 of Egr and amino acids 31–81 of Grnd. The architecture of the hetero-hexameric complex consists of a core Egr trimer bound to three Grnd-ECD copies positioned distally (Fig. 3B–D). The TNF domain of Egr adopts a typical "jelly-roll" fold, with an inner and an outer β-sheet composed of five antiparallel β-strands (conventionally named A'AHCF and B'BGDE, respectively). A single disulfide bond between the conserved cysteines, Cys353 and Cys368, links strands βE and βF, analogously to that what observed in the TNF ligands TALL1, April, and Tweak (Supplementary Fig. S3). The core Egr trimer superposes well to the structure of the *Spodoptera frugiperda* orthologue that was recently solved in the apo form (rmsd of 0.99 Å), which differs from the *Drosophila* counterpart only by the presence of a longer loop between strands βD and βE (Fig. 3A)[6]. No major conformational rearrangements of the Grnd fold can be observed upon binding to its ligand Egr (with a rmsd of 0.674 Å between the free and bound state), that associates as rigid bodies with no induced fit. Importantly, in the Egr:Grnd arrangement each copy of Grnd contacts two Egr protomers (Fig. 3B–D), implying that the trimeric state of the ligand is essential for the organisation of the high-affinity interaction surface with the receptor.

The interface between Egr and Grnd is organized in three surface patches (Fig. 3E–G). The first is centred around the β1–β2-hairpin tip of Grnd that faces two Egr subunits (Egr-1 and

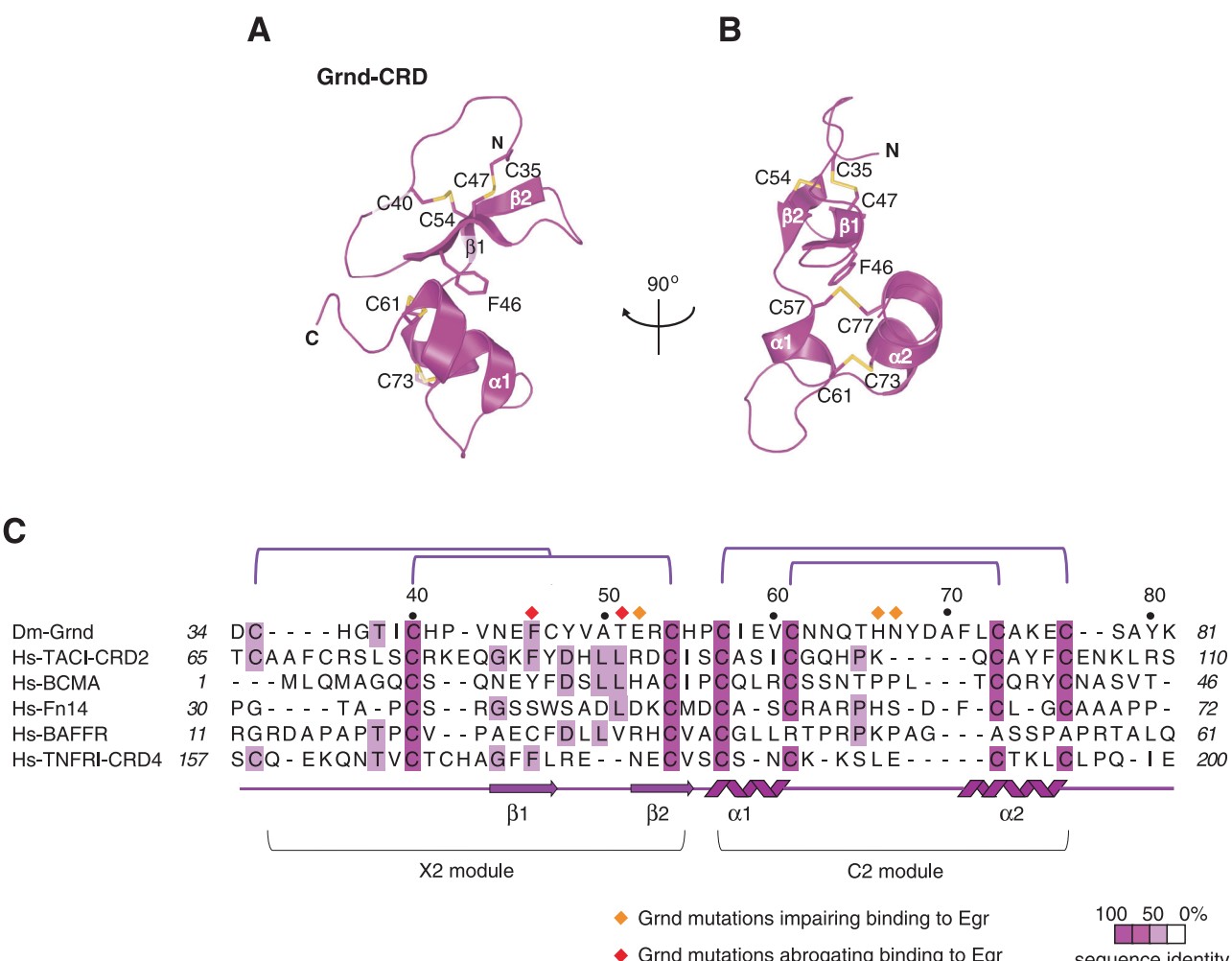

**Fig. 2 Structure of the extracellular domain of Grnd. A**, **B** Cartoon representation of Grnd-CRD viewed at the indicated orientation. Grnd is depicted in purple, with the disulfide bonds in yellow and Phe46 in sticks. **C** Structure-based sequence alignment of Grnd-CRD with CRDs of TNFRs containing X2 and C2 modules, or sharing a similar conformation with Grnd, coloured by percentage of identity. Grnd disulfide bridges are indicated as purple lines connecting cysteines numbered above the Grnd sequence. Grnd secondary structure elements are shown below the alignment. Grnd residues whose mutation impairs completely or partially binding to Egr are indicated as red or orange rhombuses, respectively.

Egr-2 in Fig. 3C, E, in yellow and orange respectively). On the hairpin, Glu52$^{Grnd}$ makes polar interactions with Arg369 of Egr-2, and the carbonyls of Val49$^{Grnd}$ and Ala50$^{Grnd}$ are hydrogen-bonded to Arg401$^{Egr-2}$. Notably, Thr51$^{Grnd}$ maintains the correct orientation of the Grnd β-hairpin by forming an intramolecular hydrogen bond with the main-chain nitrogen of Arg53$^{Grnd}$, and a hydrogen bond with Asn333$^{Egr-2}$ (Fig. 3E). The second interacting patch engages the Egr-1 protomer, and comprises Tyr48$^{Grnd}$ and Asp69$^{Grnd}$ that are hydrogen-bonded to His388$^{Egr-1}$ and Arg391$^{Egr}$ (Fig. 3F). Asp377$^{Egr-1}$ of the same Egr-1 molecule contacts His66$^{Grnd}$ and Asn67$^{Grnd}$, in a third interacting module (Fig. 3G).

To assess the relevance of these interactions for hexamer assembly, we substituted key residues identified in the structural analysis with alanines, and tested the binding ability of the mutated constructs in pull-down assays performed with Strep-Egr$^{146–409}$ immobilized on Strep-tactin beads and purified Grnd$^{30–97}$ constructs in solution. First, we asked if Grnd mutants were able to bind wild-type Egr$^{146–409}$ (Fig. 3H), and found that single substitution of Thr51Ala$^{Grnd}$ or double replacement of His66Ala-Asn67Ala$^{Grnd}$ completely abrogated the interaction between Grnd and Egr, whereas single mutations Glu52Ala$^{Grnd}$, His66Ala$^{Grnd}$ and Asn67Ala$^{Grnd}$ impaired the association only

partially. Interestingly, the Phe46Ala$^{Grnd}$ replacement also resulted in a Grnd mutant unable to interact with Egr, likely because this mutation disrupts the hydrophobic core of the receptor fold (Fig. 2A). To obtain quantitative information regarding the binding affinities of the mutants, we conducted ITC experiments, which revealed that the binding strength of the Grnd-Glu52Ala, Grnd-His66Ala and Grnd-Asn67Ala mutants for Egr$^{146–409}$ decreases about 30–100-folds compared to the nanomolar affinity of wild-type Grnd, whereas the point mutation Thr51Ala$^{Grnd}$, and the double mutation His66Ala-Asn67Ala$^{Grnd}$ completely abrogate the binding (Fig. 3I, Supplementary Fig. S4). To confirm that the Grnd mutations abrogating binding to Egr in vitro suffice to disrupt the interaction in cells, we performed co-immunoprecipitation experiments (co-IPs) using S2 cells lysate expressing HA-Egr and Flag-tagged versions of Grnd. The experiments showed that the effect of the mutations on full-length Grnd mirrors what we observed in vitro, in that His66Ala$^{Grnd}$-Asn67Ala$^{Grnd}$ and Phe46Ala$^{Grnd}$ abrogate binding, while the His66Ala$^{Grnd}$ displays a reduced affinity (Fig. 3J). We next assessed the effect of mutating Egr residues at the interface with Grnd on the ligand-receptor binding affinity in a strep pull-down analogous to the one described before. The experiment showed that the mutations Asp337Ala$^{Egr}$, His388Ala-

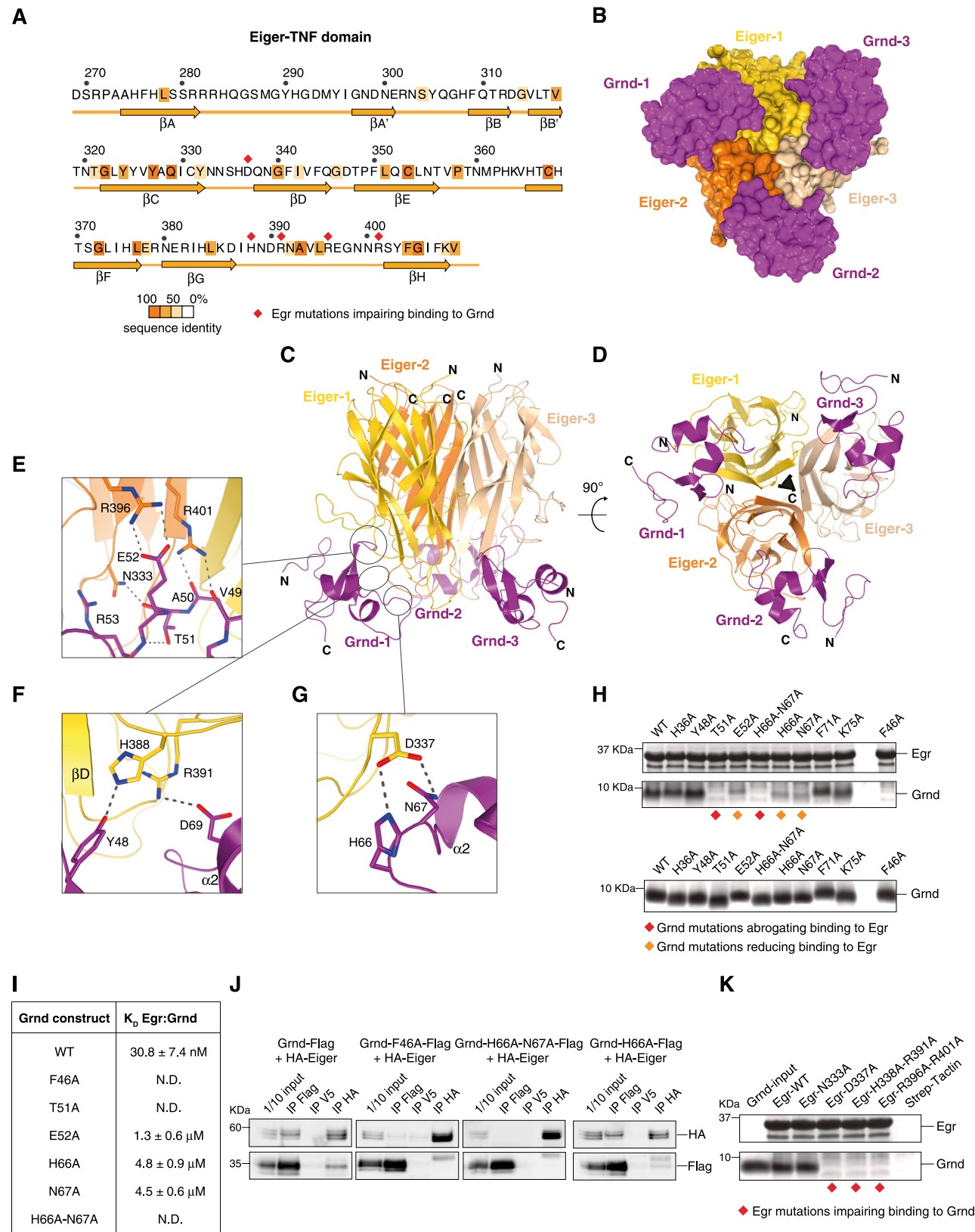

Arg391Ala[Egr] and Arg396Ala-Arg401Ala[Egr] abolish binding to Grnd (Fig. 3K), indicating that all three identified modules contribute to the binding strength between the ligand and the receptor. Most notably, the substitution of the Egr glycosylation site Asn333 with Ala does not affect the Grnd:Egr-binding affinity.

We then set out to address the relevance of the Grnd/Egr interaction in Egr-mediated signalling by introducing single or double mutations that impair binding to Egr in the endogenous *grnd* gene. We first used a combination of CRISPR-induced double-strand breaks and ends-out homologous recombination to replace the first coding exon of endogenous *grnd* with a cassette

**Fig. 3 Architecture of the Egr:Grnd complex and mapping of the interaction interface. A** Sequence of the TNF-homology domain of Egr coloured by sequence identity with TNF ligands interacting with single-CRD TNFR (see Methods for details). Residues implicated in Grnd binding are marked by red rhombuses. **B** Surface representation of the Egr:Grnd hexamer in the same orientation as in (**D**). **C, D** Cartoon representation of the Egr:Grnd complex at the indicated orientations. The three Egr-TNF copies are shown in shades of yellow, and Grnd monomers in purple. The threefold axis is indicated as a triangle. **E–G** Enlarged views of three patches of the Egr$^{269-409}$:Grnd$^{30-81}$ interface. **H** Strep pull-down assay with 2 µM Egr$^{146-409}$ absorbed on strep-tactin beads and 10 µM Grnd$^{30-97}$ in solution, either wild-type or carrying the indicated alanine substitutions, visualized by Coomassie staining. Two parts of the SDS-PAGE corresponding to the molecular weight of Egr$^{146-409}$ and Grnd$^{30-97}$ are shown. The Grnd inputs used in the pull-down experiment are shown in the bottom SDS-PAGE. Grnd mutations fully or partially impairing binding to Egr are labelled in red and orange, respectively. **I** Table of binding affinities (dissociation constants $K_D$) between Egr$^{146-409}$ and wild-type or mutated Grnd$^{30-97}$ measured by ITC. **J** Co-IP experiments between HA-tagged Eiger and Flag-tagged Grnd, wild-type or mutated as indicated, from S2 cell lysates. Bound molecules are visualized by anti-HA and anti-Flag antibodies. Immunoprecipitation of V5 was used as specificity control. **K** Strep pull-down assays analogous to the ones described in (**H**), performed with the indicated Egr$^{146-409}$ mutants and wild-type Grnd$^{30-97}$. Results of experiments presented in panels **H–J**, **K** were confirmed by successful replicate experiments. Uncropped images of SDS-PAGE gels and immunoblots are provided in Supplementary Fig. 6.

| Table 2 Data collection and refinement statistics of the Egr:Grnd structure. | |
| --- | --- |
| | **Egr:Grnd** |
| *Data collection* | ID23-2-ESRF |
| Space group | P321 |
| Cell dimensions | |
| *a, b, c* (Å) | 78.7, 78.7, 57.66 |
| *α, β, γ* (°) | 90.0, 90.0, 120.0 |
| Resolution (Å) | 57.66–2.02 (2.02–2.05)$^a$ |
| $R_{sym}$ or $R_{merge}$ | 0.21 (0.89) |
| $CC_{1/2}$ | 1.00 (0.57) |
| $I/\sigma I$ | 8.6 (2.2) |
| Completeness (%) | 99.9 (98.5) |
| Unique reflections | 13860 (633) |
| Multiplicity | 11.7 (10.1) |
| *Refinement* | |
| Resolution (Å) | 44.0–2.02 |
| $R_{work}/R_{free}$ | 0.192/0.214 |
| No. atoms | |
| Protein | 1554 |
| Water | 118 |
| *B*-factors protein/water (Å$^2$) | 37.70/42.99 |
| R.m.s. deviations | |
| Bond lengths (Å) | 0.003 |
| Bond angles (°) | 0.91 |
| Ramachandran values | |
| Favoured (%) | 95.74 |
| Allowed (%) | 4.26 |
| Outliers (%) | 0 |
| $^a$Values in parentheses are for the highest-resolution shell. | |

harbouring an *attP* reintegration site, thereby generating a *grnd* null mutant (*grnd$^{KO/KO}$*; Supplementary Fig. S5A). We then used the *attP* site in *grnd$^{KO/KO}$* null mutant animals to reinsert cDNAs encoding *wt grnd* (*grnd$^{wt}$*, control), *grnd$^{H66A,N67A}$*, *grnd$^{H66A}$*, or *grnd$^{F46A}$* at the *grnd* locus. We previously showed Grnd is required for Egr-induced apoptosis in wing imaginal discs[5]. To test the ability of the different *grnd* mutants to transduce Egr signalling in vivo, we monitored Egr-induced apoptosis in *grnd$^{KO}$/grnd$^{wt}$* (control), *grnd$^{KO}$/grnd$^{KO}$*, *grnd$^{KO}$/grnd$^{H66A,N67A}$*, *grnd$^{KO}$/grnd$^{H66A}$*, or *grnd$^{KO}$/grnd$^{F46A}$* mutant animals (Fig. 4A–F). As expected, Egr-induced ectopic wingless expression and apoptosis were abolished in *grnd* null mutant animals (*grnd$^{KO}$/grnd$^{KO}$*; Fig. 4B), but not in animals heterozygous for *wt grnd* (*grnd$^{KO}$/grnd$^{wt}$*; Fig. 4C). The double mutation His66Ala-Asn67Ala (*grnd$^{KO}$/grnd$^{H66A,N67A}$*) or the substitution Phe46Ala (*grnd$^{KO}$/grnd$^{F46}$*) completely abrogated Egr-induced apoptosis, whereas the His66Ala mutation alone (*grnd$^{KO}$/grnd$^{H66A}$*) only partially reduced Egr-mediated cell death (Fig. 4D–F). These

results indicate that direct binding of Egr to Grnd mediates Egr-dependent apoptosis, and that the high affinity of the interaction is key for the penetrance of the response.

Collectively our structural and functional analyses reveal that (i) trimeric TNF-homology domains of Egr form hetero-hexamers with the extracellular domain of Grnd, in which each receptor copy contacts two TNF ligands; (ii) the key residues at the Egr:Grnd interface are Thr51, His66 and Asn67 on Grnd and Asp337, His388-Arg391 and Arg396-Arg401 on Egr; (iii) impairment of the nanomolar binding affinity between Egr and Grnd prevents Egr-mediated caspase activation in vivo, showing that in wing discs, Grnd mediates Egr-dependent activation of JNK signalling.

**Egr colocalises with Grnd, but not Wgn, in vesicles.** The evidence that Egr directly associates with both Grnd and Wgn, although with different affinities, and that Grnd mutants impaired in Egr binding suppress Egr-induced apoptosis, open the question of how the two receptors cross-talk in transducing Egr signals. We previously showed that in wing imaginal discs the majority of Grnd enriches at the apical membrane, where it colocalises with the subapical polarity determinant Crumbs[5] (Fig. 4G, J). Conversely, Wgn is almost exclusively localised in intracellular vesicles (Fig. 4J and Supplementary Fig. S5B, B′), suggesting that in this tissue, Wgn and Grnd are implicated in distinct cellular functions. Knockdown of Wgn affects neither the level nor the localisation of Grnd (Supplementary Fig. S5B–B″). It was previously reported that Egr-induced cell death relies on its internalisation into early endosomes[20]. To visualise the localisation of Grnd in response to Egr-activation, we temporally over-expressed Egr in the wing pouch. As Egr-induced cell death leads to strong ablation of the wing pouch and delamination of dying cells (Fig. 4H), we blocked apoptosis by knocking down a downstream component of the JNK pathway, dTAK1 (Fig. 4I, K). Even in the absence of cell death, Egr overexpression results in the internalisation of Grnd in intracellular vesicles that are distinct from those immunolabeled with the anti-Wgn antibody (Fig. 4J, K, M′), but positive for Egr (Fig. 4M–M″). The observation that Egr-induced Grnd internalisation occurs in non-apoptotic discs is in accordance with previous reports showing that Egr-internalisation is required for, and not a consequence of, cell death[21,22]. Importantly, we did not observe any internalisation of Grnd$^{H66A-N67A}$ in response to Egr overexpression indicating that ligand-receptor binding is a prerequisite for Egr internalization and induced apoptosis (Fig. 4L). Strikingly, we did not observe any colocalization between internalised Egr and Wgn-positive vesicles following Egr overexpression (Fig. 4M′–M″). Altogether, our data show that Grnd and Wgn localise in separate cellular compartments and respond differently to Egr

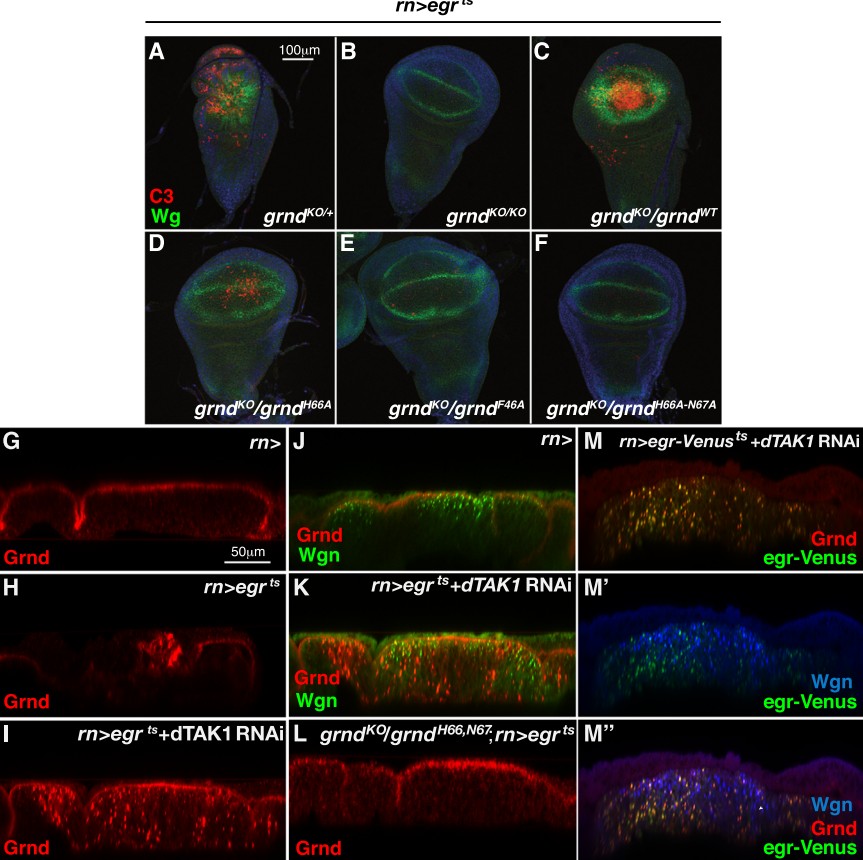

**Fig. 4 Internalisation of Egr and Grnd in vesicles that are distinct from those labelled by Wgn precedes Egr-mediated apoptosis. A–F** *Drosophila* wing imaginal discs of the indicated genotypes dissected from third instar larvae after 48 h of temperature-induced Egr expression, and stained for Wingless (Wg; green) and Caspase 3 (Cas3; red). **G–L** Transversal sections of wing discs of the indicated genotypes stained for Grnd (labelled in red) and Wgn (**J**, **K**; labelled in green). Egr mis-expression results in the internalisation of Grnd in intracellular vesicles that are distinct from those harbouring Wgn (**K**). **M–M″** Transversal sections of wing discs expressing Venus-tagged Egr stained for Grnd (labelled in red), Venus (labelled in green), and Wgn (blue) shows co-localisation of Grnd and Egr-Venus in intracellular vesicles that are distinct from Wgn-positive vesicles. Results of experiments presented in panels (**A–M**) were confirmed by successful replicate experiments.

signals, and that Egr-induced apoptotic signalling requires Grnd-mediated internalization.

## Discussion

Binding of TNF ligands to TNF surface receptors triggers signal transduction by activation of the JNK cascade. *Drosophila* code for a single TNF ligand, named Egr, and two TNFRs, Grnd and Wgn, that have been implicated in different cellular pathways[2–5]. Here we show that trimeric Egr ligands assemble in hetero-hexamers with both Grnd and Wgn, with Grnd displaying nanomolar binding affinity for Egr, which is 3-orders of magnitude higher than that for Wgn. Structural analysis reveals that the Egr:Grnd complex is organised around core Egr trimers with each Grnd copy positioned distally in the cleft between two adjacent TNF ligands. In imaginal wing discs, mutations impairing the association of Egr with Grnd reduce Egr-induced apoptosis with a penetrance reflecting the residual binding ability. Consistently, in wing discs, Egr and Grnd are internalized in vesicles, a pre-requisite for TNF-signal transduction, that are distinct from Wgn-containing vesicles.

Grnd was recently identified in a tumour suppressor screen, functionally acting as a TNFR[5]. A major finding of our structural characterization is that, in spite of the poor sequence homology, the ECD of Grnd adopts a cysteine-rich domain fold consisting of a X2 β-hairpin followed by a helical C2 module, in a combination

that has not been observed in TNFR domains so far. The reciprocal orientation of the Grnd X2-C2 modules resembles that observed in single CRDs of human TNFRs such as TACI, BCMA and Fn14 and the fourth CRD of TNFR1, with the conserved Phe46$^{Grnd}$ side-chain packing between the two (Supplementary Fig. S2). Notably, the CRD of Wgn conforms better than Grnd to known TNFRs, although it binds to Egr with much lower affinity.

*Drosophila* TNF/Egr code for a conserved TNF-homology domain that assembles in trimers as reported for all known TNFs including the recently determined *Spodoptera frugiperda* ligand[6], and induces oligomerization of the cognate TNFRs, Grnd and Wgn. Hetero-hexameric Egr:Grnd complexes organize around core ligand trimers, with Grnd molecules contacting two Egr/TNF copies with no conformational rearrangement compared to the apo receptor. The modular interaction surface is provided mainly by the Grnd β-hairpin that snugly fits in between two TNF domains and the α1-α2 loop, strengthening the binding by His66-Asn67$^{Grnd}$-mediated polar interactions. Whether pH-dependent protonation of His66$^{Grnd}$ affects the binding interaction between the ligand and the receptor in internalized endocytic compartments and the recycling remains an intriguing open question. The open arrangement of the Grnd-ECD X2-C2 modules and the rigid conformation of the β-hairpin are key in ensuring nanomolar affinity, as the Thr51Ala$^{Grnd}$ mutation disrupting the β-hairpin conformation or the Phe46Ala mutation altering the hydrophobic core of the Grnd fold completely abolish

binding. Recent findings suggested that Grnd glycosylation on Asn63[Grnd] affects Egr signalling by modulating the Egr:Grnd interaction[9]. Our structural data show that N63[Grnd] does not belong to the Grnd surface facing Egr, but rather points outwards to the hetero-hexamer. We speculate that Grnd glycosylation might affect effective TNF signalling by sterically preventing clustering of Egr:Grnd complexes at the plasma membrane. Intriguingly, the TNF domain of Egr, which faces Grnd in the complex, is also glycosylated on Asn333[Egr] (Fig. 3E). Whether Asn333[Egr] glycosylation reduces the ligand:receptor binding affinity remains an interesting open question.

An important finding of our biochemical and structural studies is that Egr associates with Grnd and Wgn with the same stoichiometry, but substantially different affinities, implying that the ligand engages with the two receptors in diverse physiological conditions. Egr is thought to represent an ancestral surveillance system that acts both locally and systemically to cope with external pathogens and tissue damage, and to rid the organism of developing tumours. Among its systemic functions, hemocyte-derived Egr has been reported to contribute to the elimination of premalignant cells in peripheral tissues through Grnd-mediated apoptosis. More recently, Agrawal et al reported another systemic function of Egr in coupling growth with nutrient availability[8]. In response to nutritional deprivation, fat-derived Egr inhibits growth by blocking the release of insulin-like peptides from the insulin-producing cells (IPCs) in the brain. This function of Egr depends on Grnd, whose expression pattern in the brain is restricted to the IPCs. Tissues exposed to distant sources of Egr are likely to experience much lower concentrations than those exposed to locally produced Egr. This might explain why Grnd, which has a much higher affinity for Egr, and not Wgn, is required for Egr-mediated systemic responses. In line with this, local, but not systemic, sources of Egr can activate Wgn in nociceptive sensory neurons to increase the nociceptive (pain) sensitivity response following UV-induced epidermal damage[23,24]. Interestingly, several recent studies describe a function for Wgn in the central nervous system, which is also where Egr is most highly expressed[25]. Thus, Keller et al showed that during the process of neuromuscular junction degeneration, glial-derived Egr triggers a degenerative response by signalling through Wgn in neighbouring motoneurons. A third study found that Wgn function in neurons to increase sleep in response to Astrocyte-derived Egr[26]. Common to all of the above examples is that Wgn activation in neurons relies on a local source of Egr, which could reflect a relative low affinity of Wgn for its ligand and/or a preference for membrane-bound TNF (mTNF). In mammals, TNFR1 binds to both soluble TNF (sTNF) and mTNF with high affinity, while TNFR2 preferentially binds mTNF and has poor affinity for sTNF[27]. One could speculate that differential affinities of TNFRs for soluble TNF could serve as a way of decoupling local (neuronal) and systemic stress responses. Our studies show that while Grnd is localised at the membrane in imaginal discs, Wgn is primarily found in intracellular vesicles, suggesting that in this tissue, Grnd and Wgn are activated by different mechanisms. One study reported that RNAi-mediated knockdown of Wgn suppresses the Egr-induced cell death in the eye compound[4]. By contrast, we did not observe suppression of Egr-induced apoptosis neither in *wgn* null mutant animals, nor upon RNAi-mediated depletion[5]. Here we show that overexpression of Egr in wing imaginal discs results in internalisation of Grnd and Egr in vesicles that are distinct from Wgn-positive vesicles providing additional evidence that Grnd is the *bona fide* receptor mediating Egr-dependent apoptosis in imaginal tissues. The observation that the majority of Wgn is localised in intracellular vesicles, and therefore not directly accessible to binding of extracellular Egr, poses the question as to how Wgn might be

activated. Different scenarios for Wgn activation could be imagined including re-localisation of Wgn to the membrane in response to specific stimuli and/or ligand-independent receptor activation. Interestingly, Ruan et al reported that Wgn, but not Egr, is required for targeting of photoreceptor axons, providing one example of a physiological process where Wgn signals independently of Egr[21]. Future studies with mutants impairing the Egr:Wgn interaction are needed to elucidate the molecular mechanisms of Grnd-mediated and Wgn-mediated Egr activities, and to dissect the cross-talk between the two receptors. Most importantly, the concept of threshold-dependent activation of different TNFRs by the same TNF ligand uncovered in *Drosophila* by our studies opens the question as to whether similar working principles pertain human TNFRs, which we anticipate will be a fascinating direction for future investigations in the field.

## Methods

**Protein expression and purification**. The *Drosophila* Grnd extracellular domain, corresponding to residues 30–97, was cloned in a pETM14 vector with a cleavable N-terminal His$_6$-tag, and expressed in SHuffle B cells (NEB) by induction with 0.5 mM IPTG at 16 °C overnight. Cells were lysed in 0.1 M Tris-HCl pH 8, 0.3 M NaCl, and 10% glycerol, and centrifuged for 1 h at 100,000 × *g*. The clear lysate was affinity-purified by injection on a HiTrap chelating column (GE Healthcare) loaded with NiCl$_2$. The protein was eluted by applying a 5–200 mM imidazole gradient, and dialysed overnight against 0.01 M Tris-HCl pH 8.0, 0.03 M NaCl and 5% glycerol in the presence of PreScission protease to remove the His$_6$-tag. After injection into a 6-ml Resource-Q anion exchange column, Grnd$^{30–97}$ was eluted with a gradient of 30–120 mM NaCl in 10 column volumes. Peak fractions containing pure protein were pooled, desalted in 0.01 M Hepes pH 7.5 and 0.05 M NaCl, and stored at −80 °C.

The Egr extracellular domain encompassing residues 146–409 was cloned in a pET43 vector with an N-terminal His$_6$-tag and expressed in SHuffle cells similarly to Grnd$^{30–97}$. Clear cell lysates were affinity-purified on a HiTrap column. Peak fractions eluting from the HiTrap column were pooled, concentrated and injected on a HiLoad 16/600 Superdex-200 column (GE Healthcare) equilibrated in a buffer containing 0.02 M Tris pH 8.0, 0.3 M NaCl, and 5% glycerol. All purification steps were conducted at 4 °C. For strep-binding assays, Egr$^{146–409}$ was cloned into a pET100 vector, and purified by affinity on strep-tactin beads (IBA-Lifescience). Grnd and Egr point mutations were generated by QuikChange strategy (Agilent) according to manufacturer's instructions. All clones were sequence verified.

*Drosophila* Wgn$^{78–201}$, comprising the whole extracellular domain, was cloned in a pETM14 vector, and expressed in SHuffle B cells. After lysis and centrifugation, the clear lysate was affinity-purified on a HiTrap chelating column. The eluted protein was dialysed overnight against 0.01 M Tris-HCl pH 8.0, 0.04 M NaCl and 5% glycerol in the presence of PreScission protease to remove the His$_6$-tag. Wgn$^{78–201}$ was further purified on a 6-ml Resource-Q anion exchange column. Proteins-containing fractions were pooled, and stored at −80 °C. All primers used for the cloning are listed in Supplementary Table 1.

**Protein crystallization and structure determination**. To obtain diffraction-quality crystals, Grnd$^{30–97}$ was incubated with trypsin in a 1:50 weight:weight ratio for 16 h at 4 °C, and subsequently purified on a HiLoad 16/600 Superdex-200 column equilibrated in 0.02 M Tris-HCl pH 8.0 and 0.05 M NaCl. Initial crystallization trials of trimmed Grnd$^{30–81}$ were performed with the commercial screen Index HT and Crystal Screen HT (Hampton Research) at 82 mg/ml in sitting-drop vapour diffusion using a Mosquito nanodispenser (TTP Labtech), mixing 100 nl protein solution with an equal volume of reservoir at 20 °C. Crystals appeared after 2 days in about 30% of conditions containing PEG or MPD. Crystals were flash-cooled in liquid nitrogen without additional cryo-protection. The best diffracting crystals of Grnd$^{30–81}$ grew with a reservoir containing 0.1 M Hepes pH 7.5, 45% MPD, and 0.2 M ammonium acetate. X-ray diffraction data were collected to 0.926 Å resolution at beamline PXIII of the Swiss Light Source (SLS, Switzerland) at 0.978 Å wavelength. Crystals belong to the space group $P4_12_12$, with a single molecule per asymmetric unit. The structure was determined using sulfur single-wavelength anomalous diffraction (S-SAD) experimental phases exploiting the high content of sulfur-containing residues of Grnd$^{30–81}$. High-multiplicity S-SAD data to 1.92 Å resolution were collected at various χ angles of the PRIGo goniometer[28] at a wavelength of 2.075 Å at the SLS beamline X06DA-PXIII as described elsewhere[29]. Data were processed with XDS[30], using the automated toolbox xia2[31]. Initial phases and model were derived using the SHELX-suite programme[32] in hkl2map[33]. The initial model built from the S-SAD dataset was used for molecular replacement in the 0.926 Å dataset, and progressively optimized by iterative cycles of manual building in Coot[34] and refinement in Phenix[35]. To refine anisotropically atomic displacement parameters, the last rounds of refinement were carried out with SHELX till the final $R_{free}$ of 15.6% and $R_{work}$ of 13%. The final model contains residues 34–81 of Grnd. Data statistics are shown in Table 1.

To obtain crystals of the ligand/receptor complex, purified Egr[146–409] and Grnd[30–97] were incubated in a 1:1.3 molar ratio for 45 min at 4 °C, and then subjected to limited proteolysis by addition of trypsin in a 1:50 weight:weight ratio. After 2 h at 4 °C, the proteolyzed complex was purified on a Superdex-200 10/300 SEC column (GE Healthcare) equilibrated in 0.02 M Hepes pH 7.5, 0.3 M NaCl, 5% glycerol. Peak fractions were pooled and concentrated to 28 mg/ml. The boundaries of the trimmed constructs were identified by a combination of intact MW determination by MALDI-TOF MS at the MS/proteomics facility (Cogentech S.c.a. r.l, Milan, Italy), and N-terminal sequencing (AltaBioscience, Redditch, UK). Initial crystals of the trimmed complex appeared in a few hours at 20 °C in the Pre-Crystallization Test (PCT, Hampton Research) upon manual mixing 1 μl of the proteolyzed Egr[269–409]:Grnd[30–81] sample with 1 μl of 0.1 M Tris-HCl pH 8.5, 0.2 M MgCl₂, 15% PEG 4000. After optimization, well-diffracting crystals grew by hanging-drop vapour diffusion with a reservoir containing 0.1 M Tris-HCl pH 8.5, 14% PEG 4000, 0.2 M MgCl₂. Crystals were cryo-protected by adding 20% ethylene glycol to the reservoir condition, and flash-cooled in liquid nitrogen. X-ray diffraction data to 2.02 Å resolution were collected at the beamline ID23-2 of the European Synchrotron Radiation Facility (Grenoble, France) at a 0.873 Å wavelength. Data were processed with XDS[30]. Crystals belong to the space group P321, with one copy of the trimmed Egr[269–409]:Grnd[30–81] complex per asymmetric unit. The biologically relevant 3:3 hetero-hexamer presented in Fig. 3 was generated by applying the crystallographic threefold symmetry to the 1:1 Egr[269–409]:Grnd[30–81] complex present in the asymmetric unit. The structure was solved by molecular replacement with Phaser[36], using as a search model the pdb obtained with Phyre2[37] threading the Egr sequence on the coordinates of human EDA-A2 (PDB-ID: 1RJ8). The model of Grnd[30–81] previously determined was then fitted manually in the electron density obtained by molecular replacement, and the Egr:Grnd complex was progressively optimized by interactive cycles of manual building in Coot[34], and refinement in Phenix[36]. The final model was refined to R_free of 21.4% and R_work of 19.2%, and includes residues 31–81 of Grnd and residues 269–409 of Egr. Data statistics are given in Table 2. All structures were illustrated with PyMol (DeLano Scientific LLC).

**Pull-down assay**. To analyse the Egr:Grnd interface in Fig. 3H–K, 2 μM of strep-tagged Egr[146–409] were adsorbed on Strep-tactin beads and incubated for 1 h at 4 °C with 10 μM of Grnd[30–97], in a buffer consisting of 0.01 M Hepes pH 7.5, 0.1 M NaCl, 5% glycerol, 0.05% Tween, 0.5 mM EDTA. After washing, species retained on beads were separated on SDS-PAGE, and visualized by Coomassie staining.

**Analytical size-exclusion chromatography (SEC)**. For analytical SEC of Fig. 1B, C, 30 μM or 300 μM of Egr[146–409] were loaded singularly or in combination with Grnd[30–97] or Wgn[78–201] on a Superdex-5/150 column (GE Healthcare) equilibrated in 0.01 M Hepes pH 7.5, 0.3 M NaCl, and 5% glycerol, and eluted in 50 μl fractions. For Fig. S1A, Egr[146–409] and Wgn[78–201] were loaded singularly or mixed at 30 μM equimolar concentration. In Fig. S3B, 30 μM of Egr[146–409] were incubated with Grnd[30–97] in a 1:1.3 molar ratio for 45 min at 4 °C. After treatment with trypsin, the proteolyzed complex was loaded on a Superdex-5/150 column. Peak fractions were analysed by SDS-PAGE followed by Coomassie staining.

**Isothermal titration calorimetry (ITC)**. To measure the binding affinity of the Egr:Grnd and Egr:Wgn interaction, purified Egr[146–409], Grnd[30–97] (either wild-type or mutated in key residues), and Wgn[78–201] were dialysed overnight against 0.02 M Tris-HCl pH 8.0 and 0.3 M NaCl. The thermodynamical parameters of their association were measured using a MicroCal PEAQ-ITC (Malvern, UK) at 25 °C. For the Egr:Grnd interaction, Grnd[30–97] was titrated at 100–340 μM into 8 μM Egr[146–409] with 19 μl injections, while for the Egr:Wgn binding Wgn[78–201] was titrated at 4.5 mM into 410 μM Egr[146–409] with 13 μl injections. Heats of ligand binding were fitted to a single-site binding curve with the MicroCal PEAQ-ITC software.

**Sequence alignment**. The extracellular portion of the Wgn sequence from Drosophila melanogaster (Uniprot entry Q9VWS4) was aligned to the TNF receptors Homo sapiens TACI (Uniprot entry O14836), Mus musculus TACI (Uniprot entry Q9ET35), Gallus gallus TACI (Uniprot entry A4Q912), Danio rerio TACI (NCBI Reference Sequence XP_017213984.1), Homo sapiens BCMA (Uniprot entry Q02223), Mus musculus BCMA (Uniprot entry O88472), Falco peregrinus BCMA (NCBI Reference Sequence XP_027647601.1), Xenopus laevis BCMA (NCBI Reference Sequence XP_018092389.1), Homo sapiens Fn14 (Uniprot entry Q9NP84), Mus musculus Fn14 (Uniprot entry Q9D8D0), Xenous laevis Fn14 (Uniprot entry Q6SIX7), Homo sapiens BAFFR (Uniprot entry Q96RJ3), Mus musculus BAFFR (Uniprot entry Q96RJ3), Gallus gallus BAFFR (Uniprot entry Q3bk46) with Clustal Omega[38]. The same receptors were aligned with the extra-cellular portion of the Grnd sequence from Drosophila melanogaster (Uniprot entry Q9VJ83), with the addition of Homo sapiens TNFRI (Uniprot entry P19438), Mus musculus TNFRI (Uniprot entry P25118), Manacus vitellinus TNFRI (Uniprot entry A0A093PSM9), Xenopus laevis TNFRI (Uniprot entry A9CPG4), and Danio rerio TNFRI (Uniprot entry F1QQS0) via Clustal Omega. The Drosophila melanogaster Egr sequence (Uniprot entry Q8MUJ1) was aligned to the TNF ligands Homo sapiens TALL1 (Uniprot entry Q9Y275), Mus musculus TALL1 (Uniprot

entry Q9WU72), Gallus gallus TALL1 (Uniprot entry Q8JHJ4), Xenopus laevis TALL1 (Uniprot entry R9U5E4), Danio rerio TALL1 (Uniprot entry F1QUA2), Homo sapiens April (Uniprot entry O75888), Mus musculus April (Uniprot entry Q9D777), Xenopus laevis April (Uniprot entry L7T0Q5), Danio rerio April (Uniprot entry D2DMJ8), Homo sapiens Tweak (Uniprot entry O43508), Mus musculus Tweak (Uniprot entry 054907), Xenopus laevis Tweak (Uniprot entry), and Danio rerio Tweak (NCBI Reference Sequence XP_018095209.1) via Clustal Omega.

**Static light scattering**. In Fig. 1D, static light scattering (SLS) analyses of Grnd[30–97] and Wgn[78–201] were performed on a Viscotek GPCmax/TDA (Malvern, UK) instrument, connected in tandem with a series of two TSKgel G3000PWxl size-exclusion chromatography columns (Tosoh Bioscience). The system was equilibrated in a buffer containing 0.01 M Hepes pH 7.5 and 0.05 M NaCl, and calibrated with BSA. SLS analyses of Egr[146–409], and the complexes Egr[146–409]:Grnd[30–97] and Egr[146–409]:Wgn[78–201] were conducted coupling a Superdex 200-10/300 increase column to the same Viscotek GPCmax instrument equilibrated in a buffer containing 0.02 M Tris-HCl pH 8.0, 0.3 M NaCl, 5% glycerol. Typically, about 75 μL of the samples concentrated at about 2 mg/mL were loaded on the columns, and eluted isocratically.

**Fly strains and food**. Animals were reared at 25 °C on fly food containing 17 g inactivated yeast powder, 83 g cornflour, 10 g agar, 60 g white sugar and 4.6 g Nipagin M (in ethanol) per liter. The rn-Gal4, UAS-egr, tubGAL80ts animals were raised at 18 °C, shifted to 30 °C for 40 h during early third instar larval development (7 days AEL), and subjected to dissection immediately after[39].

**Cell culture**. Drosophila S2 cells were grown in complete Schneiders medium (CSM; Schneiders medium (Invitrogen) supplemented with 10% heat-inactivated FBS (BioWhittaker), 100 U/ml penicillin and 100 μg/ml streptomycin (Invitrogen) at 25 °C. Transfections were done using Effectene reagent (Qiagen).

**Plasmids**. grnd coding sequences were PCR-amplified from BDGP EST complementary DNA clones RE28509, and cloned into the pENTR/D-TOPO vector using the following gene-specific primers: sense primer CACCATGTCGGTCAG-GAAGTTGAG; antisense primer GAAAGCGACGGGAATCGTCGC. The mutations giving rise to the single F46A and H66A and double H66A- N67A amino acid changes in Grnd were introduced using the QuickChange site-directed mutagenesis kit (Agilent) and the following primers: for the F46A mutation, sense 5′-CTGTC ATCCGGTCAATGAAGCATGCTATGTTGCAACGGAGAG-3′, 5′-antisense CTCTCCGGTTGCAACATAGCATGCTTCATTGACCGGATGACAG-3′; for the H66A mutation, sense 5′-GGTCTGCAATAATCAAACCGCCAACTACGAT GCGTTTCTG-3′, antisense 5′-CAGAAACGCATCGTAGTTGGCGGTTTGATT ATTGCAGACC-3′; for the H66A-N67A mutation, sense 5′-TCTGCAATAAT CAAACCGCCGCCTACGATGCGTTTCTGTGCGC-3′, antisense 5′-GCGCACA GAAACGCATCGTAGGCGGCGGTTTGATTATTGCAGA-3′.

To generate Grnd-WT-Flag, Grnd-F46A-Flag, grnd-H66A-Flag and grnd-H66A-N67A-Flag, grnd, grndF46A, grnd-H66A and grnd-H66A-N67A coding sequences were cloned into the Gateway Destination vector pActin-33Flag (Drosophila Gateway Vector Collection) for expression in S2 cells. The HA-Egr plasmid was a gift from M. Miura (Department of Genetics, Graduate School of Pharmaceutical Sciences, The University of Tokyo, Japan).

**Generation of grnd^WT and grnd^KO, grnd^H66A, grnd^F46A and grnd^H66A,N67A mutant flies**. To generate a grnd null mutant, an approach combining the CRISPR technique and homologous recombination was used as described in ref. [40]. Double-strand breaks were induced by the CRISPR technique using single-stranded guide (sg)RNAs and a Cas9-encoding plasmid. For optimal targeting of the grnd locus, sgRNA target sequences were selected as 20-nt sequences preceding an NGG PAM sequence in the genome (GN20GG). To generate pCFD4{grnd^KO}, gRNAs targeting sequences immediately before and after exon I in the grnd locus were cloned into the tandem gRNA expression vector, pCFD4 (kind gift from Simon Bullock, MRC- Laboratory of Molecular Biology, Cambridge, UK (Addgene plasmid # 49411), using the grnd-5′-KO-pCFD4-FORWARD: 5′-TAT ATA GGA AAG ATA TCC GGG TGA ACT TCg agc gtc tgg gcc gcg tta cGT TTT AGA GCT AGA AAT AGC AAG-3′ and grnd-3′-KO-pCFD4-REVERSE: 5′-ATT TTA ACT TGC TAT TTC TAG CTC TAA AAC tga act tgc ata gaa ccc gcc GAC GTT AAA TTG AAA ATA GGT C-3′ (grnd-specific sequences are in lower case) primers as described in http://www.crisprflydesign.org/wp-content/uploads/2014/06/Cloning-with-pCFD4. pdf. For homologous recombination, two homology arms were amplified from genomic DNA using the following primers: for homology arm I, sense: 5′-GCG GCC GCC GTA TAG TTC ATA TTG GGA TAC TGG GAA TTT C-3′ and antisense: 5′-GGT ACC GCG GCC CAG ACG CTA AC-3′, for homology arm II, sense: 5′-ACT AGT AGG TTT TTC CAG CTG GAC TTA ATT G-3′ and antisense: 5′-GGC GCG CCC ATA ACC GTT GTG GGC GTG G-3′ (in bold are the added restrictions sites used for cloning into pTV2). The resulting PCR products were digested and cloned into the pTV2 vector (kind gift from Cyril Alexandre and Jean-Paul Vincent, the Francis Crick Institute, London, UK) to generate pTV2{grnd^KO, mini-white}. To facilitate homologous recombination, embryos were injected with pTV2{grnd^KO, mini-white} in the presence of pCFD4{grnd^KO} and a

Cas9-containing plasmid. After confirmed targeting, the resulting strain harbours a deletion of the entire first codon exon in the *grnd* locus and is referred to as *grnd*[KO] in the manuscript. The *grnd*[KO] strain harbouring an *attP* integration site, was used as a host for reintegration of cDNAs encoding either *grnd*[WT], *grnd*[F46A], *grnd*[H66A] or *grnd*[H66A-N67A]. For this purpose, *grnd*[WT], *grnd*[F46A], *grnd*[H66A] or *grnd*[H66A-N67A] were PCR-amplified from plasmids and cloned into the reintegration vector RIV[cherry] (kind gift from Cyrille Alexandre and Jean-Paul Vincent, the Francis Crick Institute, London, UK) giving rise to RIV[cherry]{*grnd*[WT], pax-RFP}, RIV[cherry]{*grndF46A*, pax-RFP}, RIV[cherry]{ *grnd-H66A*, pax-RFP} and RIV[cherry]{ *grnd-H66A-N67A*, pax-RFP}. Reintegration was achieved by co-injecting a *grnd*[KO] strain expressing the PhiC31 integrase.

**Transgenic flies**. The *w*[1118]; +; *rn*-Gal4, *UAS-Egr*, *tubGal80ts/TM6B* line was provided by I. Hariharan, Dept of Molecular & Cell Biology, University of California, Berkeley, USA. The *UAS-egr-venus* line was a gift from Marcos Vidal, Beatson Institute for Cancer Research, Glasgow, UK. The *en-Gal4, UAS-GFP/Cyo* line was provided by Bruce A. Edgar, Huntsman Cancer Institute, Salt Lake City, USA. The *rn*-Gal4 (BL7405) line was provided by the Bloomington Drosophila Stock Center. The following RNAi lines were from the collections of the Vienna Drosophila RNAi Center (VDRC): *tak1* RNAi (KK101357) and *wgn* RNAi (GD9152).

**Antibodies**. The following primary antibodies were used for immunofluorescence in this study: guinea pig anti-Grnd (1/500) and mouse anti-Wgn (1/100) described in ref. [5], mouse anti-Wg (#4D4) from DSHB 1/50, rabbit anti-cleaved caspase-3 (Asp 175) from Cell Signaling Technology #9661 1/500.

**Immunostainings of larval tissues**. Wing imaginal discs dissected from late third instar larvae in 1×PBS (0.137 M NaCl, 2.7 mM KCl, 4.3 mM Na$_2$HPO$_4$, 1.47 mM KH$_2$PO$_4$, pH 8) were fixed in 4% formaldehyde (Sigma) in PBS for 20 min at room temperature washed in PBS containing 0.1% Triton X-100 (PBT), blocked for 2 h in PBT containing 10% FBS (PBS-TF), and incubated overnight with primary antibodies at 4 °C. The next day, cells and tissues were washed, blocked in PBS-TF, and incubated with secondary antibodies at 1/500 dilution for all of them (Cy3-conjugated donkey anti-rabbit, Cy3-conjugated donkey anti-mouse and Cy3-conjugated mouse anti-guinea pig, and Cy5-conjugated donkey anti-mouse from Jackson ImmunoResearch; goat anti-mouse Alexa Fluor 488 from Invitrogen) for 2 h at room temperature. After washing, tissues were mounted in Vectashield (Vector Labs) or Prolong (Invitrogen) containing DAPI for staining of DNA. Fluorescence images were acquired using a Leica SP5 or SP8 and processed using Adobe Photoshop.

**Lysis and immunoprecipitations**. Cells were lysed in 100 ml buffer A (0.05 M Tris-HCl (pH 8), 0.15 M NaCl, 1% NP-40, 1 mM EGTA, 0.5 M NaF, phosphatase inhibitor cocktail 2 (Sigma), complete protease inhibitor cocktail (Roche)). Immunoprecipitations were performed from about 1 × 10$^7$ S2 cells. Cells were lysed in 200 μl buffer A, and cell extracts were cleared of membranous material by centrifugation at 10,000 r.p.m. for 15 min. The cleared extracts were incubated with protein G-Sepharose beads (Sigma) for 1 h to reduce unspecific binding of proteins to the beads in the subsequent purifications. Next, the precleared extracts were incubated with 80 μl of protein G-Sepharose beads and 1 μl of the relevant antibody for 3 h. Subsequently, beads were washed three times in buffer A, boiled in the sample and reducing buffers (Invitrogen).

**Western blotting**. Proteins were resolved by SDS-PAGE using 4–12% gradient gels (NuPAGE Novex gel, Invitrogen) and transferred electrophoretically to polyvinylidene difluoride membranes (Milipore). The membranes were incubated for 1 h in blocking buffer (PBS, 5% milk) and incubated overnight at 4 °C in the same buffer containing primary antibodies at 1:1000 dilutions: mouse anti-Flag (Sigma #F3165), rat anti-HA (Roche #3F10). Membranes were washed three times in PBS-T, blocked for 1 h, and probed with secondary antibodies in blocking buffer for 1 h at room temperature. After three washes in PBS-T, chemiluminescence was observed using the ECL-Plus western blotting detection system (Amersham Biosciences). Images were generated using the Fujifilm Multi Gauge software.

**Genotypes**.
Figure 4:
A: *w*[-]; *grnd*[KO]/+; *rn*-Gal4, *UAS-egr*, *tubGal80ts*/+
B: *w*[-]; *grnd*[KO/KO]; *rn*-Gal4, *UAS-egr*, *tubGal80ts*/+
C: *w*[-]; *grnd*[KO/WT]; *rn*-Gal4, *UAS-egr*, *tubGal80ts*/+
D: *w*[-]; *grnd*[KO/H66A]; *rn*-Gal4, *UAS-egr*, *tubGal80ts*/+
E: *w*[-]; *grnd*[KO/F46A]; *rn*-Gal4, *UAS-egr*, *tubGal80ts*/+
F: *w*[-]; *grnd*[KO/H66A-N67A]; *rn*-Gal4, *UAS-egr*, *tubGal80ts*/+
G: *w*[-]; +; *rn*-Gal4/+
H: *w*[-]; +; *rn*-Gal4, *UAS-egr*, *tubGal80ts*/+
I, K: *w*[-]; *UAS-tak RNAi*/+; *rn*-Gal4, *UAS-egr*, *tubGal80ts*/+
J: *w*[-]; +; *rn*-Gal4/+
L: *grnd*[KO/H66A-N67A]; *rn*-Gal4, *tubGal80ts*/+

M–M″: *w*; *UAS-tak RNAi*/+; *rn*-Gal4, *UAS-egr*, *tubGal80ts*/ *UAS-egr*-venus
Supplementary Fig. 4:
B-B″: *w*; *en-Gal4, UAS-GFP*/+; *UAS-wgn RNAi*/+

**Reporting summary**. Further information on research design is available in the Nature Research Reporting Summary linked to this article.

## Data availability
Data supporting findings of this manuscript are available from the corresponding authors upon reasonable request. Coordinate and structure factors have been deposited to the Protein Data Bank under the accession code PDB-ID 6ZSY for Grnd-ECD native, 6ZSZ for Grnd-ECD S-SAD, and 6ZT0 for the Grnd:Egr complex. Uncropped images of SDS-PAGE gels and immunoblots are provided in Supplementary Fig. 6.

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

## Acknowledgements
We thank the scientists at the X06DA beamline at the Swiss Light Source, at the Diamond beamlines, and at the European Synchrotron Radiation Facility beamlines for valuable support in data collection. We thank the Imaging Platform and Jutta Bulkescher at DanStem. We are grateful to all members of the Mapelli, Colombani and Andersen laboratories for scientific discussion and for carefully reading the manuscript. V.P. was PhD student within the European School of Medicine (SEMM) and recipient of an AIRC fellowship (2015–2018). This work was supported by a grant to M.M. from the Italian Association for Cancer Research (AIRC) (IG 18629) and the Ministry of Health (RF.2013-02357254). This was partially supported by the Italian Ministry of Health with Ricerca Corrente and 5x1000 funds. J.C. and D.S.A. are funded by H2020 European Research Council grant number 803630 and Novo Nordisk Foundation grant number NNF18OC0033920. The Novo Nordisk Foundation Center for Stem Cell Biology is supported by a Novo Nordisk Foundation grant number NNF17CC0027852.

## Author contributions
V.P. performed the biochemical experiments; V.P. and S.M. purified the proteins; V.P., V.C. and S.P. crystallized the proteins; S.P. and V.O. collected and processed the diffraction data; S.P. solved and refined the structures; Q.L., R.L., J.C. and D.S.A. generated Grnd mutants flies and performed the in vivo experiments; S.P., J.C., D.S.A. and M.M. supervised the project and wrote the manuscript.

## Competing interests
The authors declare no competing interests.
