## [Peer Review File · Nature Communications]

REVIEWER COMMENTS

Reviewer #1 (Remarks to the Author):

The Drosophila genome includes one TNF ligand (Eiger) and two receptors: Grindelwald and Wengen. Mapelli and colleagues nicely combine in this paper biochemistry, structural biology, cell biology and genetics to demonstrate, on one hand, that Drosophila TNF (Eiger) binds with higher affinity to Grindelwald than to Wengen and that increase levels of TNF changes the subcellular localization of Grindelwald but not Wengen. On the other hand, authors solve the structure of the Eiger/Grindelwald complex and carry out functional experiments in vitro and in Drosophila to identify the relevant amino acids mediating the interaction. I have no major concern with the paper and I think it is a good candidate for Nature Communications. Figures are elegant, paper is well written and discussed and the topic is highly relevant in the field of signaling and TNF biology. I have to admit, though, that I am a geneticist and not very familiar with structural biology studies. So, figures 2 and 3 should be better reviewed by a specialist. I have only the following minor comment:

- Scale bars should be included in Figure 4 and its supplementary figure(s)
- Eiger and Wengen no-colocalization should be monitored with co-stainings in the same wing disc expressing Eiger-Venus (if possible)
- Is Eiger internalized in the absence of Grindelwald? Authors show that mutant Grindelwald is not upon Eiger overexpression, but about Eiger internalization in this mutant background. Authors might want to use their new mutants to analyze this and reinforce the Eiger/Grindelwald functional relationship

Reviewer #2 (Remarks to the Author):

In this study Palmerini et al., present a detailed structural analysis of the interactions between the Drosophila TNF Eiger and its receptors Grindelwald and Wengen. The authors biochemically characterize the components of the corresponding complexes and their binding strengths. To enhance crystallizability, the authors trim their protein constructs via limited proteolysis, a powerful method to improve chances of obtaining crystals. The crystallographic datasets have sound statistics with good CC1/2 values in the highest resolution shells, and the refined models possess good geometry. The structural data is thoroughly described and validated by a substantial panel of mutations. Finally, the authors demonstrate functionally the importance of Eiger interactions for induction of apoptosis in Drosophila.

The manuscript is clearly written and the data is nicely presented. The Drosophila TNF system is an important model for the more complex mammalian counterparts, and as such a structural description of it should be a useful and welcome addition to the field. Besides a likely technical problem with the presentation of the SEC data, I only have a few minor comments.

Comments:

The elution volumes in Fig. 1B-C are not congruent with descriptions in the text and the molecular weight estimates from SEC are dubious. In line 136 the authors state that the Egr:Grnd complex elutes at a volume corresponding to 120 kDa, referring to Fig. 1B. However, the Egr:Grnd peak in the SEC profile elutes significantly before the 158 kDa standard. Similarly, in the same figure, Egr (presumably a trimer of around 90 kDa) runs at an elution volume corresponding to the 158 kDa standard. Fig. 1C has the same problems. Can the authors explain these large discrepancies, especially in light of very accurate and believable molecular weights from SLS, presented in Fig. 1D? Could it be that the authors misplotted the SEC standard and it is shifted vs the samples? Please also check the supplemental figures accordingly. Can the authors also please plot the SEC starting from an elution

volume of 0ml, instead of 1ml?

Minor comments:

1) Could the authors please showcase sample density for the obtained structures in the supplement? Especially in the case of the 0.93Å Grnd data, sample density would go a long way in presenting data quality.

2) line 176: It would be useful here to note RMSD values of the overall closest structural homologues, such as obtained via the DALI server

3) lines 193-195: Does this mean that the full complex is generated by applying the crystal symmetry? How many copies of each protein are in the asymmetric unit? One or two sentences of clarification would be welcome here.

4) Histidines constitute pH switches in many systems and change protonation going from neutral to late endosomal pH. Given the presence of His residues at the Egr:Grnd interface and the internalisation pathway, can the authors comment on the possible relevance of protonation change?

5) the label for Grnd is not aligned with the band in the bottom gel of Fig.1B

Max Renner
University of Oxford

Reviewer #3 (Remarks to the Author):

Drosophila is a nice system to study the TNF pathway. So far, only a single TNF-family ligand (Eiger) has been identified and one or two TNF receptors. In previous work done by Colombani and Andersen, Grindelwald was identified as the most important receptor for Eiger to mediate apoptosis. This was a key development because, prior to that it was assumed that Wengen was the Eiger receptor. In this study, the authors show compelling information linking Eiger with Grindelwald. They provide a structure of the two extracellular domains interacting and then use mutagenesis to show that changes that would be predicted to disrupt these interactions have the expected properties in vivo. A surprising finding was that a residue that was thought to be necessary for N-linked glycosylation and might regulate ligand-receptor affinity was not on the binding interface. The work on Grindelwald is well done and well presented.

In contrast, there is no evidence from this study that Wengen functions as an Eiger receptor in vivo, it has binding affinities in the micromolar range. The authors cite work from that suggests that Wengen might function as a receptor when Eiger is presented locally (or could there be modifications necessary for binding).

Overall this is an important piece of work. No additional experiments are necessary. However, the authors should discuss the possibility that at least this form of Wengen may not be a functional Eiger receptor in vivo.

Reviewers' comments:

Reviewer #1 (Remarks to the Author):

The Drosophila genome includes one TNF ligand (Eiger) and two receptors: Grindelwald and Wengen. Mapelli and colleagues nicely combine in this paper biochemistry, structural biology, cell biology and genetics to demonstrate, on one hand, that Drosophila TNF (Eiger) binds with higher affinity to Grindelwald than to Wengen and that increase levels of TNF changes the subcellular localization of Grindelwald but not Wengen. ON the other hand, authors solve the structure of the

Eiger/Grindelwald complex and carry out functional experiments in vitro and in Drosophila to identify the relevant amino acids mediating the interaction. I have no major concern with the paper and I think it is a good candidate for Nature Communications. Figures are elegant, paper is well written and discussed and the topic is highly relevant in the field of signaling and TNF biology. I have to admit, though, that I am a geneticist and not very familiar with structural biology studies. So, figures 2 and 3 should be better reviewed by a specialist. I have only the following minor comment:

We thank the reviewer for the positive and constructive comments on the manuscript. We are delighted that the manuscript was well received.

- Scale bars should be included in Figure 4 and its supplementary figure(s)

Scale bars have now been added to Figure 4 and its supplementary figure.

- Eiger and Wengen no-colocalization should be monitored with co-stainings in the same wing disc expressing Eiger-Venus (if possible)

We have now added new images in figure 4 (M-M'') showing a co-staining of Eiger-Venus, Wengen, and Grindelwald in the same wing disc. The image clearly illustrates a co-localization of Grnd and Eiger-Venus in vesicles that are distinct from Wgn positive vesicles and replaces panel M-M'' in the previous figure 4M (showing co-localisation between grnd and egr-venus) and figure S5C-C'' (showing non-overlap between Grnd and Wgn). No co-localization between Eiger-Venus and Wgn is observed.

- Is Eiger internalized in the absence of Grindelwald? Authors show that mutant Grindelwald is not upon Eiger overexpression, but about Eiger internalization in this mutant background. Authors might want to use their new mutants to analyze this and reinforce the Eiger/Grindelwald functional relationship.

We attempted to do address this point and observed that although apoptosis is blocked in *grnd* null mutant discs overexpressing Egr – as shown in the attached figure 1 for the Reviewer below – Egr, which is produced by the tissue itself, is found in vesicles. As Egr cannot bind its receptor, it is not trapped at the membrane. Since the apoptosis induced by Egr is rescued in the *grnd* null mutant background, Egr positive vesicles most likely reflects trafficking of Egr produced by the tissue. We have chosen not to include this figure in the manuscript as it does not add to the data already presented in figure 4.

Figure 1 for Reviewer

A-B''- XY sections of wing imaginal disc overexpressing *egr-Venus* (green) in the pouch of animal heterozygous (control, A-A'') or homozygous for a *grnd* null mutation (B-B'') showing a rescue of the ablation of the wing pouch and an absence of *grnd* staining (red) in animals mutant for *grnd* (B-B''). B''' - Transversal section of *grnd* mutant discs shows that although ablation of the discs is blocked, trafficking of Egr-positive vesicles is observed.

Reviewer #2 (Remarks to the Author):

In this study Palmerini et al., present a detailed structural analysis of the interactions between the *Drosophila* TNF Eiger and its receptors Grindelwald and Wengen. The authors biochemically characterize the components of the corresponding complexes and their binding strengths. To enhance crystallizability, the authors trim their protein constructs via limited proteolysis, a powerful method to improve chances of obtaining crystals. The crystallographic datasets have sound statistics with good CC1/2 values in the highest resolution shells, and the refined models possess good geometry. The structural data is thoroughly described and validated by a substantial panel of mutations. Finally, the authors demonstrate functionally the importance of Eiger interactions for induction of apoptosis in *Drosophila*.

The manuscript is clearly written and the data is nicely presented. The *Drosophila* TNF system is an important model for the more complex mammalian counterparts, and as such a structural description of it should be a useful and welcome addition to the field. Besides a likely technical problem with the presentation of the SEC data, I only have a few minor comments.

We thank the Reviewer for the positive comments on the overall quality and significance of the work, we are delighted the manuscript has been well-received. We have addressed the points raised as described below.

Comments:

The elution volumes in Fig.1B-C are not congruent with descriptions in the text and the molecular weight estimates from SEC are dubious. In line 136 the authors state that the Egr:Grnd complex elutes at a volume corresponding to 120 kDa, referring to Fig. 1B. However, the Egr:Grnd peak in the SEC profile elutes significantly before the 158 kDa standard. Similarly, in the same figure, Egr (presumably a trimer of around 90 kDa) runs at an elution volume corresponding to the 158 kDa standard. Fig.1C has the same problems. Can the authors explain these large discrepancies, especially in light of very accurate and believable molecular weights from SLS, presented in Fig.1D? Could it be that the authors misplotted the SEC standard and it is shifted vs the samples? Please also check the supplemental figures accordingly. Can the authors also please plot the SEC starting from an elution volume of 0ml, instead of 1ml?

We thank the Referee for the comments. The sentence in line 136 describing the SEC elution profile displayed in Fig. 1B is misleading as it refers to the molecular weight of the Egr:Grnd complex measured by SLS experiments, as detailed in Fig. 1D (i.e. 116 KDa), rather than to the size-exclusion-chromatography runs. In fact, the Egr:Grnd complex elutes from a Superdex-200 column at the same volume of the 158 KDa molecular-weight-marker (Mwm), slightly earlier than what observed for Egr in isolation. We know that the Egr construct used for these analyses, spanning residues 146-409, forms trimers of 90 KDa (Fig. 1D) but elutes from a Superdex-200 column around the 158 KDa Mwm (Fig. 1B-C, Supplementary Fig. S1A), likely because it consists of a globular TNF-Homology-Domain preceded by a 124-residue long unstructured region that might affect the run. To make sure that there were no frameshift problems between the Egr; Egr:Grnd and the Mwm elution profiles plotted in Figure 1, we have re-run the Egr sample used in the experiments of Fig. 1 (see enclosed *Figure 2 for the Reviewer*). This experiment confirmed that the elution profiles reported in Figure 1 and Supplementary Figure 1 are correct.

Figure 2 for Reviewer. SEC elution profile of Egr_144-409. Comparison of the elution profiles of Egr_146-409 samples from a Superdex-200 SEC column. The profile presented in Figure 1 of the submitted manuscript (orange trace) is overlaid with a new run of the same sample (brown trace) and the Mwm run (gray dotted trace). Both Egr samples elute similarly to the 158 KDa Mwm.

We have corrected the sentence at line 136 according to these considerations, and replotted the SEC profiles in panel 1B-C and S1A from 0.0 ml, as suggested by the Referee.

Minor comments:

1) Could the authors please showcase sample density for the obtained structures in the supplement? Especially in the case of the 0.93Å Grnd data, sample density would go a long way in presenting data quality.

We thank the Reviewer for the suggestion. We have included two panels illustrating the quality of the electron density map of Grnd (Supplementary Figure S2B) and the Egr:Grnd complex (Supplementary Figure S3C).

2) line 176: It would be useful here to note RMSD values of the overall closest structural homologues, such as obtained via the DALI server

We agree with the Reviewer that a direct structural comparison of the Grnd-ECD fold with other TNFR domains would be informative on the putative evolutionary pathways of the TNFR family. An initial Dali search with the newly determined Grnd-ECD fold did not yield any homologous domain. Thus, we decided to directly compare Grnd-ECD with the fold of structurally-characterized TNFR extracellular domains consisting of a single Cysteine-Rich-Domain, i.e. TACI, BCMA and Fn14, already depicted in Supplementary Figure S2D-E-F (S2C-D-E in the first submission). To this aim, we first used the Rapido server (<http://rapido.embl-hamburg.de/>) to identify the largest overlapping rigid bodies between Grnd-ECD and the selected TNFR domains, and then used this information to superpose the TNFRs. This procedure revealed that Grnd/TACI superpose with an RMSD of 5.2 Å, Grnd/BCMA and Grnd/Fn14 superpose with an RMSD of 4.4 Å. We have added this information to the main text and illustrated the superposition result in a Supplementary Fig. 2G and corresponding legend.

3) lines 193-195: Does this mean that the full complex is generated by applying the crystal symmetry? How many copies of each protein are in the asymmetric unit? One or two sentences of clarification would be welcome here.

The Reviewer is correct in saying that the Grnd:Egr hetero-hexamers are generated by applying the 3-fold crystallographic symmetry to the single copy of the Grnd:Egr 1:1 complex present in the asymmetric unit. We have specified this evidence in the Methods section.

4) Histidines constitute pH switches in many systems and change protonation going from neutral to late endosomal pH. Given the presence of His residues at the Egr:Grnd interface and the internalisation pathway, can the authors comment on the possible relevance of protonation change?

We thank the Referee for the comment. Indeed, in the case of EGFR and its ligands (human EGF, mouse EGF and TGF α) it has been reported that the pH of the endosomal compartments (pH around 6 for early endosomes) affects the affinity between ligands and receptor, this way impacting on the recycling or degradation fate of the internalized receptor/ligand complex (Maeda et al, J Control Release 2002, doi: 10.1016/s0168-3659(02)00126-8). Specifically, in this study the authors show that the pH-sensitive affinity between the EGFR and the ligands partly depends on histidine residues present on the ligand surface. Whether the Egr/Grnd interaction is also modulated by pH-sensitive histidines at the Egr:Grnd interface, including Grnd-H66 and Egr-H338, remains to be tested. Because very little is known on the internalization process of Egr with Grnd and Wengen, we have added a short sentence in the discussion about these considerations, leaving open a thorough evaluation of the question for *ad hoc* future studies.

5) the label for Grnd is not aligned with the band in the bottom gel of Fig.1B

We thank the Reviewer for having noticed this mislabeling. We have corrected panel 1B moving the Grnd label in line with the Grnd band of the corresponding SDS-PAGE.

Max Renner
University of Oxford

Reviewer #3 (Remarks to the Author):

Drosophila is a nice system to study the TNF pathway. So far, only a single TNF-family ligand (Eiger) has been identified and one or two TNF receptors. In previous work done by Colombani and Andersen, Grindelwald was identified as the most important receptor for Eiger to mediate apoptosis. This was a key development because, prior to that it was assumed that Wengen was the Eiger receptor. In this study, the authors show compelling information linking Eiger with Grindelwald. They provide a structure of the two extracellular domains interacting and then use mutagenesis to show that changes that would be predicted to disrupt these interactions have the expected properties *in vivo*. A surprising finding was that a residue that was thought to be necessary for N-linked glycosylation and might regulate ligand-receptor affinity was not on the binding interface. The work on Grindelwald is well done and well presented.

In contrast, there is no evidence from this study that Wengen functions as an Eiger receptor *in vivo*. It has binding affinities in the micromolar range. The authors cite work from that suggests that Wengen might function as a receptor when Eiger is presented locally (or could there be modifications necessary for binding).

Overall this is an important piece of work. No additional experiments are necessary. However, the authors should discuss the possibility that at least this form of Wengen may not be a functional Eiger receptor *in vivo*.

We are grateful to the Reviewer for the flattering revision, and we are pleased to see that our work is so nicely received. Based on our biochemical data, indicating a very low binding affinity between Wengen and Egr, and the functional data showing that Wengen cannot be internalized in response to high Egr concentrations, it would be tempting to speculate that Wengen is not a functional receptor, at least in the wing discs. However, we cannot exclude that a Wengen-mediated Egr internalization occurs in the presence of different genetic lesions or stimuli, or in other tissues. Another point that might explain their differential activation by the same ligand is the observation that mammalian TNFR1 binds to both soluble and membrane-bound TNF (sTNF and mTNF) with high affinity, while TNFR2 preferentially binds mTNF and has poor affinity for sTNF (Grell et al. 1995, Cell) (We have included this point in the discussion). Hence, although Wgn is clearly not required for Egr-induced apoptosis in the disc, we cannot exclude that Wgn, like TNFR2, display higher affinity for mTNF, and contribute to other TNF-dependent processes not included in our studies. For these reasons, we would prefer not to push forward the idea that Wengen is not a functional receptor.

REVIEWERS' COMMENTS

Reviewer #1 (Remarks to the Author):

Authors have satisfactorily addressed all my concerns and, as such, I recommend this nice piece of work for publication in Nature Communications.

Reviewer #2 (Remarks to the Author):

The authors state in their rebuttal that "...likely because it consists of a globular TNF-Homology-Domain preceded by a 124-residue long unstructured region that might affect the run."

I suggest, for further clarification, the authors mention in the manuscript their hypothesis that the disordered region may be affecting the hydrodynamic properties and thus elution volume.

Otherwise, the authors have addressed my concerns sufficiently and I recommend acceptance of the manuscript.

Reviewer #3 (Remarks to the Author):

I am happy with the revised manuscript. No additional changes necessary. I recommend acceptance.

REVIEWERS' COMMENTS

Reviewer #1 (Remarks to the Author):

Authors have satisfactorily addressed all my concerns and, as such, I recommend this nice piece of work for publication in Nature Communications.

We are pleased to know that the Reviewer is satisfied by the revision, and we thank him/her for the constructive comments during the review process.

Reviewer #2 (Remarks to the Author):

The authors state in their rebuttal that "...likely because it consists of a globular TNF-Homology-Domain preceded by a 124-residue long unstructured region that might affect the run."

I suggest, for further clarification, the authors mention in the manuscript their hypothesis that the disordered region may be affecting the hydrodynamic properties and thus elution volume.

Otherwise, the authors have addressed my concerns sufficiently and I recommend acceptance of the manuscript.

We thank the Reviewer for the comment. As suggested, we have included a sentence in the main text that clarifies how the disordered N-terminal fragment of Egr1¹⁴⁶⁻⁴⁰⁹ might affect the hydrodynamic properties of the construct, and hence the SEC elution volume.

Reviewer #3 (Remarks to the Author):

I am happy with the revised manuscript. No additional changes necessary. I recommend acceptance.

We are glad to know that the Reviewer is happy with the revised version of the manuscript, and we seize the occasion to thank him/her for all the positive comments.